# DUX Hunting—Clinical Features and Diagnostic Challenges Associated with *DUX4*-Rearranged Leukaemia

**DOI:** 10.3390/cancers12102815

**Published:** 2020-09-30

**Authors:** Jacqueline A. Rehn, Matthew J. O’Connor, Deborah L. White, David T. Yeung

**Affiliations:** 1Cancer Program, Precision Medicine Theme, South Australian Health & Medical Research Institute (SAHMRI), Adelaide, SA 5000, Australia; jacqueline.rehn@sahmri.com (J.A.R.); david.yeung@adelaide.edu.au (D.T.Y.); 2Faculty of Health and Medical Science, University of Adelaide, Adelaide, SA 5000, Australia; 3Michael Rice Centre for Haematology and Oncology, Womens’s & Children’s Hospital, North Adelaide, SA 5006, Australia; 4Australian Genomics, The Murdoch Children’s Research Institute, Parkville, VIC 3052, Australia; 5Australian and New Zealand Children’s Oncology Group (ANZCHOG), Clayton, VIC 3168, Australia; 6Department of Haematology, Royal Adelaide Hospital, Adelaide, SA 5000, Australia

**Keywords:** acute lymphoblastic leukaemia, *DUX4*, *ERG*, molecular subtype

## Abstract

**Simple Summary:**

*DUX4*-rearrangement (*DUX4r*) is a recently discovered recurrent genomic lesion reported in 4–7% of childhood B cell acute lymphoblastic leukaemia (B-ALL) cases. This subtype has favourable outcomes, especially in children and adolescents treated with intensive chemotherapy. The fusion most commonly links the hypervariable *IGH* gene to *DUX4* a gene located within the D4Z4 macrosatellite repeat on chromosome 4. *DUX4r* is cryptic to most standard diagnostic techniques, and difficult to identify even with next generation sequencing assays. This review summarises the clinical features and molecular genetics of *DUX4*r B-ALL and proposes prospective new diagnostic methods.

**Abstract:**

*DUX4*-rearrangement (*DUX4r*) is a recently discovered recurrent genomic lesion reported in 4–7% of childhood B cell acute lymphoblastic leukaemia (B-ALL) cases. This subtype has favourable outcomes, especially in children and adolescents treated with intensive chemotherapy. The fusion most commonly links the hypervariable *IGH* gene to *DUX4* a gene located within the D4Z4 macrosatellite repeat on chromosome 4, with a homologous polymorphic repeat on chromosome 10. *DUX4r* is cryptic to most standard diagnostic techniques, and difficult to identify even with next generation sequencing assays. This review summarises the clinical features and molecular genetics of *DUX4*r B-ALL and proposes prospective new diagnostic methods.

## 1. Introduction

B-cell acute lymphoblastic leukaemia (B-ALL) is a malignant disorder of the bone marrow resulting in over proliferation of immature B lymphoblasts. The disease can manifest at any age but the majority of patients are children, making B-ALL the most common childhood malignancy [1]. This heterogeneous disease is characterised by a variety of different genomic alterations including changes in chromosome number, chromosomal translocations and single nucleotide variants (SNV). Detection of the underlying genomic alterations assists clinical risk stratification and therapeutic triage. Cytogenetic analysis has proven adept at identifying several recurrent genomic alterations which result in diseases with distinct gene expression profiles (GEP) and defined prognosis. This includes high hyperdiploidy, hypodiploidy, and the translocations t(12;21) [*ETV6-RUNX1*], t(9;22) *BCR-ABL1*, t(1;19) *TCF3-PBX1* and alterations of chromosome 11q23 resulting in rearrangement of *KMT2A/MLL*. These alterations account for approximately 60% of pediatric B-ALL cases [2,3]. Remaining patients were historically classified as *B-other* and demonstrated highly variable prognosis and treatment response.

Molecular studies involving GEP and next-generation sequencing (NGS) have subsequently identified a number of additional recurrent molecular alterations not detectable with standard cytogenetics, several of which may be targetable by precision medicine approaches. This includes the newly recognized subtype of Philadelphia chromosome-like (Ph-like) ALL characterized by a gene expression profile similar to cases with a *BCR-ABL1* translocation, but instead carrying one of multiple kinase activating lesions. One example is rearrangement of the cytokine receptor gene *CRLF2* (*CRLF2*r), commonly with concurrence of Janus Kinase 2 (*JAK2*) mutations [4,5], affecting 5–7% of children with B-ALL [3]. NGS has also recently identified a rearrangement of the homeodomain encoding the Double Homeobox 4 (*DUX4*) transcription factor with the immunoglobulin heavy chain (*IGH*) locus which results in a distinct genetic subtype.

As early as 2002 researchers identified a novel B-ALL subtype, with a distinct microarray GEP, not associated with any known recurrent genomic alterations, that appeared to confer a good prognosis. Interrogation of overexpressed genes in these patients failed to uncover a causative lesion [6]. Follow-up studies involving copy number alteration (CNA) analysis revealed many of the patients with the distinct expression profile also demonstrated deletion of the *ERG* gene (ETS transcription factor *ERG*), a genomic alteration absent in almost all other subtypes. *ERG* deletion was consequently proposed as the driving lesion in this subtype [7,8]. Multiple studies have subsequently demonstrated that monoallelic deletion of *ERG* is observed in only a subset of patients demonstrating this GEP. Furthermore, *ERG* deletions were subclonal in several patients at diagnosis and either altered or absent at relapse [9,10,11].

In 2016, two independent studies identified rearrangement of the *DUX4* locus (*DUX4*r), most commonly partnered with *IGH*, present in patients with the previously detected GEP [12,13]. Transduction of the *DUX4* fusion transcript into NIH3T3 fibroblasts resulted in cellular transformations, demonstrating the oncogenic potential of this alteration [12]. Multiple studies have subsequently confirmed the unique GEP of *DUX4*r cases and identified the rearrangement in 4–7% of B-ALL patients [9,14,15,16,17,18,19], as well as within the NALM6 cell line [12]. Leukaemic cells carrying the *DUX4*r also display a unique methylation profile, associated with widespread hypomethylation [15,20,21], and express a specific non coding RNA signature [16,21]. Consequently, *DUX4*r, also reported in the literature as *DUX4/ERG*, has increasingly been accepted as a distinct molecular subtype in B-ALL. In this review, we present the current understanding of the molecular structure and biological effects of the *DUX4*r. We further explore the continued difficulties associated with detection of this alteration in new patient samples and discuss the impact this may have on the accurate assessment of prognosis and subsequent therapy options.

## 2. Description of the DUX4 Rearrangement

The *DUX4* gene is present within each repeat of the D4Z4 tandem array located in the subtelomeric region of chromosome 4q [22], with an almost identical locus (>98% nucleotide identity) on 10q [23,24]. The D4Z4 array is polymorphic in length containing between 11–100 copies of the 3.3 kb repeat in healthy adults (Figure 1) [25]. In healthy tissue, transcription of *DUX4* is restricted to germline cells of the testes. Transcription has also been observed in induced pluripotent stem cells, suggesting a role for *DUX4* in germline development [26,27]. Expression of the full-length *DUX4* transcript is epigenetically silenced in somatic tissue [27]. Only the first exon of the spliced transcript contains a coding sequence for the protein which consists of two N-terminal homeodomains capable of DNA binding [28] and a C-terminal transactivation domain [29]. The DUX4 protein is capable of binding to, and upregulating expression of, multiple genes as well as initiating expression from alternate promoters, producing non-canonical transcript isoforms [29].

Contraction of the D4Z4 region resulting in fewer than 10 repeats is associated with facioscapulohumeral muscular dystrophy (FSHD) [25,30], a genetically inherited disorder that initially manifests as progressive weakening of the facial, shoulder and upper arm muscles [31]. Partial deletion of the D4Z4 array is associated with hypomethylation and loss of repressive histone modifications that are believed to reduce chromatin packing of the subtelomeric region allowing *DUX4* expression [32,33]. Intriguingly, FSHD only manifests in patients who demonstrate D4Z4 contraction on chromosome 4q and not the homologous array on chromosome 10q. Furthermore, contraction of an alternative chromosome 4q allele (4qB) [24], does not result in disease. Sequencing efforts have subsequently revealed that the permissive 4qA allele carries a polymorphism in the region immediately distal to the final repeat of the array (Figure 1). This polymorphism creates a canonical polyadenylation signal in the 3’UTR of *DUX4* enabling expression of a stable mRNA transcript [34]. The translated protein then binds multiple target genes resulting in widespread changes in gene expression that are ultimately cytotoxic [26].

*DUX4* has also been implicated in several cancers involving rearrangements that produce chimeric proteins with altered transcriptional activity [35,36]. For example, a recurrent translocation between Capicua Transcriptional Repressor (*CIC*) and *DUX4* occurs in a proportion of patients with Ewing-like sarcoma. This chromosomal rearrangement produces an in-frame transcript containing the first 20 exons of *CIC* but replacing the terminal exons with the 3’ portion of *DUX4*. Translation of the chimeric transcript produces a protein that retains the majority of CIC, including the N-terminal DNA binding domains, but replaces the C-terminus with the DUX4 transactivation domain. As a result, the chimeric CIC-DUX4 protein acts as an oncogenic transcriptional activator [37]. In B-ALL, translocation of *DUX4* results in a different chimeric protein, but one that again acts as an oncogenic transcriptional activator. In all but one reported case, the 5’ coding sequence of *DUX4* is cryptically inserted into an alternate genomic location, resulting in expression of a chimeric transcript which retains sequence containing the N-terminus of *DUX4* but replacing the 3’ coding sequence (Figure 2). While multiple potential fusion partners have been identified, including *ERG*, *DUX4* is most commonly inserted into the *IGH* locus [9,12,13,17].

Multiple rearrangements of *IGH* (*IGH*r) have been reported in B-ALL resulting in expression or overexpression of genes with oncogenic potential. The most common of these is a translocation between chromosome 14 and *CRLF2* of the pseudoautosomal region of chromosome X/Y resulting in increased expression of cytokine receptor-like factor 2 [4,38]. In the case of *DUX4*, most analyses report that the rearrangement involves a portion of the D4Z4 array on 4q or the homologous region on 10q, consisting of either a partial copy of *DUX4* or one complete and one partial D4Z4 repeat, being inserted into the *IGH* locus, placing them close to the *IGH* enhancer (E) [12]. As with other *IGH*r, the presence of the enhancer induces expression of the translocated gene [38]. Repeats containing *DUX4* can be inserted in either orientation resulting in expression from the positive or negative strand. In some cases, a more complex rearrangement involving sequences from a third genomic location have been reported (Figure 1C) [12,15]. Alternatively, Hi-C data performed on the NALM6 cell line suggest that a reciprocal translocation can occur in which the telomeric ends of 4/10q are exchanged with 14q [39].

*IGH* breakpoints are enriched in the 3.5 kb region preceding the *IGHM* constant allele and overlapping the *IGH D-J* junctions but can occur throughout the locus [9]. Breakpoint locations for the *DUX4* gene are harder to define given the repetitive nature of the D4Z4 array, but most commonly occur within the 5’ region upstream of *DUX4* and within the 3’ coding region of exon1. This results in a *DUX4* transcript which maintains the homeodomains encoded at the 5’ end of the transcript fused to sequence, usually from *IGH-JH* or *IGH-DH* regions, but can also be another genomic location [9,12,13,40]. The resulting protein thus maintains its ability to bind *DUX4* targets but possesses a truncated C-terminus with inclusions of some amino acids encoded by the alternate locus. As the genomic breakpoints for this rearrangement are highly variable, the resultant length and amino acid sequence the C-terminal domain of the DUX4-fusion varies considerably between patients but consistently retains the DNA binding homeodomains (Figure 2) [9,12,39].

## 3. Disease Model

*DUX4*r B-ALL results in high-level expression of *DUX4* [9,12]. Paradoxically, over-expression of wild-type *DUX4* (WT DUX4) in a variety of cell types results in apoptosis [12,41]. To reconcile this paradox Yasuda et al. [12] suggest that the altered C-terminus of *DUX4* which occurs due to the cryptic insertion results in a variant of the DUX4 transcription factor that is capable of binding DNA and altering transcription but does not lead to apoptosis. Follow up experiments conducted by Tanaka et al [42] support this hypothesis. Chromatin immunoprecipitation followed by sequencing (ChIP-seq) performed on the NALM6 cell line confirmed that DUX4-IGH binds to 97% of the same gene targets as WT DUX4. However, experiments involving inducible vectors led to up-regulation of significantly fewer genes by DUX4-IGH than WT DUX4. Additionally, luciferase reporter assays utilising the ZSCAN4 promoter, a well-known target of DUX4, indicated that DUX4-IGH demonstrates decreased transcriptional activity compared to DUX4 [42]. Similar experiments conducted with Reh cells transfected with patient-derived constructs of the *DUX4-IGH* fusion confirm this finding [43] and suggest that alteration of the C-terminus of DUX4-IGH attenuates the transcription inducing ability compared to WT DUX4.

The C-terminal transactivation domain of WT DUX4 associates with p300/CBP which act as co-activators of transcription [29]. Loss of the C-terminus is therefore predicted to prevent recruitment of p300/CBP and reduce the ability of DUX4-IGH to upregulate expression. This hypothesis is supported by studies showing clear differences between the GEP generated in cells transduced to express a full-length *DUX4* transcript (*DUX4*-fl) and those expressing a truncated form of *DUX4* (*DUX4*-s) lacking the C-terminal transactivation domain (Figure 2) [26,44]. Induced expression of *DUX4-fl* results in upregulated expression of several hundred genes, many of which normally function in germline or early stem cells. Only a subset of these genes is upregulated in cells expressing C-terminally truncated forms of *DUX4*. Furthermore, expression of *DUX4-s* does not result in cell death [26]. Additional research is needed, however, both to understand how dysregulated expression induced by DUX4-IGH results in leukaemic transformation and to determine how patient specific variation in the C-terminal portion of DUX4-IGH affects disease progression.

## 4. Genomic Landscape of *DUX4*r Leukaemia 

Multiple genomic alterations are highly associated with the *DUX4*r subtype, the most prevalent of which is deletion or alteration of *ERG*. *ERG* is a transcription factor from the ETS family involved in the regulation of hematopoietic stem cell maintenance and differentiation [45,46]. Monoallelic intragenic deletions of *ERG*, most commonly involving deletion of exons 3–7 or exons 3–9, is observed in 3–5% of B-ALL cases, and almost exclusively in *DUX4*r [47,48]. *ERG* deletion was initially proposed to be the key mechanism for ERG deregulation [7], but is absent in 20–40% of patients with *DUX4*r. Furthermore, these deletions are often subclonal and the deletion breakpoint can vary between diagnosis and relapse indicating that *ERG* deletion is a secondary event in disease progression [10,47].

*DUX4*r patients have also been shown to express an alternative transcript isoform of *ERG* in which transcription is initiated at a non-canonical exon present in intron 6. This alternate *ERG* transcript (*ERG*alt) produces a truncated C-terminal ERG protein with an N-terminal encoded by 7 amino acids from the non-canonical exon 6, that acts as a competitive inhibitor of wild-type ERG [9]. DUX4-IGH has been shown to bind to an alternative transcription initiation site of within intron 6 of ERG, inducing expression of ERGalt [9]. Efforts to correlate the presence of *ERG* deletion with the expression of *ERG*alt show that levels of the *ERG*alt transcript are higher in patients with a detected monoallelic deletion [10]. Given that these deletions remove intron 6 where transcription of the alternative isoform is initiated, this increased expression must be occurring on the non-deleted allele, potentially resulting in further impairment of wild-type *ERG* transcription [9]. The role of *ERG*alt in disease progression, however, remains to be fully elucidated.

Patients with *ERG* deletion, even those with subclonal deletions, demonstrate good prognosis [10,47]. *ERG* has been shown in vivo and in mouse models to increase cell proliferation and is believed to be crucial for maintenance of human leukaemia [45]. In acute myeloid leukaemia (AML), duplication of ERG is significantly related with higher levels of expression and lower overall survival [49] while high ERG expression has been reported as a poor risk indicator in T-cell acute lymphoblastic leukaemia (T-ALL) [50]. Deletion or inhibition of *ERG* in B-ALL may therefore attenuate this effect. Cell line experiments investigating the role of *ERG* in leukaemogenesis have shown that short hairpin RNA (shRNA) knockdown of wild-type *ERG* led to inhibition of cell growth [45]. Together this data supports the observation that *ERG* deregulation offers a protective effect that may result in improved patient outcome.

Deletion of the transcription factor IKAROS Family Zinc Finger 1 (*IKZF1*), particularly the intragenic exon 4–7 deletion producing a dominant-negative isoform of Ikaros, have also been observed in a proportion of patients with *DUX4*r [8,19,47,48]. *IKZF1* deletion has been reported in a variety of B-ALL subtypes but occurs with greater frequency in patients with the *BCR-ABL1* translocation or Ph-like subtype [4,51]. Studies assessing the effect of co-occurring copy number alterations have demonstrated that *IKZF1* deletion is an independent indicator of poor prognosis [52,53] except when observed in combination with ERG deletion [54]. This may explain why the presence of *IKZF1* deletion does not translate into poorer outcomes for *DUX4*r patients [47,48]. Additional genomic alterations reported in *DUX4*r include deletion, or less commonly mutation of the lymphoid transcription factor, *PAX5*, and deletion of the cell cycle regulator *CDKN2A* or the paralogous gene *CDKN2B* [8,9,16,19]. Mutations in other transcriptional regulators not commonly affected in B-ALL, such as *MYC*, *MYCBP2*, and *ZEB2* are less frequently observed, while mutations affecting Ras signaling, as well as epigenetic modifiers *KMT2D* [15], *SETD2* and *NCOR1* have also been reported [9,18]. Given that *DUX4*r is itself capable of inducing oncogenesis the role and significance of these additional mutations is as yet unclear.

## 5. Detecting *DUX4*r 

Early research predominantly relied on the detection of intragenic *ERG* deletion as a marker for *DUX4*r [47], using Multiplex ligation-dependent probe amplification (MLPA) [55], SNP array [8,47] or genomic PCR using primers specific for the suspected deletion [10]. Genomic PCR showed higher sensitivity than MLPA [10] but neither was able to detect all cases with *DUX4*r as indicated by GEP [9,10,47]. While previously used as a surrogate of *DUX4*r, *ERG* deletion is now recognized as a secondary alteration and no longer relied upon for the detection of *DUX4*r. The expression of the *ERG*alt transcript produced in the presence of the DUX4-chimeric protein has also been suggested as a surrogate marker for *DUX4*r [9]. However, studies have shown that *ER*Galt can be expressed, albeit at lower levels, in other B-ALL subtypes. Furthermore, *ERG*alt expression may not be detected in all *DUX4*r patient samples and thus does not constitute an accurate or sensitive marker for subtype determination [10,15].

Diagnostic techniques for the detection of chromosomal translocations typically involve cytogenetic analysis, including G-banded karyotyping, fluorescence in situ hybridisation (FISH), or reverse transcriptase polymerase chain reaction (RT-PCR) for the detection of gene fusions with well-characterised breakpoints [56]. However, all of these techniques fail to conclusively detect fusions involving *DUX4*. *DUX4*r are cytogenetically cryptic due to the small size of the repeat sequence that is inserted into the *IGH* locus or [9], in the case of a reciprocal translocation, the fact that both *DUX4* and *IGH* are located in subtelomeric regions [39]. Even though FISH has been highly effective at detecting other ALL specific translocations undetectable by karyotyping [57,58], the design of fluorescent probes to indicate the *DUX4*r is difficult for two reasons. Firstly, the repetitive nature of the D4Z4 array means that probes binding to this location will bind to multiple loci both on 4q and 10q. They could potentially cross-hybridize with additional highly homologous regions present on the short arms of all acrocentric and pericentromeric regions of multiple other chromosomes [59]. Secondly, the highly variable sequence and positions of breakpoints both within *IGH* and *DUX4* make the design of break-apart fluorescent probes difficult. 

Immunophenotyping has been employed for the prediction of some subtype defining genomic lesions in ALL [60]. Whilst surface expression of CRLF2 lends itself to flow cytometric detection of patients with *CRLF2*r [61], the detection of *DUX4*r is reliant upon expression of the cell surface antigen CD371 (CLL-1). This antigen is encoded by *CLEC12A*, known to be upregulated by *DUX4* expression, and was detected on blast cells of *DUX4*r cases but shown to be almost absent in all other analysed subtypes in one study [62]. The few non-*DUX4*r samples with CD371 expression were shown to contain alternate subtype-specific lesions detectable by standard cytogenetics. Aberrant expression of CD2 has also been observed in patients with *DUX4*r [7,55] including patients who demonstrate monocytic lineage switch [55,63] and the combination of CD371 and CD2 has recently been suggested as a strong marker for *DUX4*r [60,62]. Additional studies are needed to confirm this finding and validate immunophenotyping as an accurate method for detection of *DUX4*r patients.

To date detection of the *DUX4*r subtype has been dependent almost exclusively upon NGS. Whole genome sequencing (WGS) has proven effective at detecting *DUX4*r, including determination of the breakpoint locations [13], but given its high cost is infrequently employed and not amenable as a diagnostic technique. Long read sequencing has been successfully performed on the NALM6 cell line characterised by the *DUX4-IGH* rearrangement in order to clarify genomic breakpoints and determine the full-length fusion transcript [12,39]. However, this form of sequencing has not been applied outside of research. More recently, genomic capture high throughput sequencing (gc-HTS) was utilised in a research setting for detection of clonal rearrangement of the IGH/TR-δ loci used for MRD monitoring. This sequencing approach was not only effective at detecting IGH and TRD rearrangements but also detected the presence of rearrangements involving the *IGH* locus (*IGH*r) including 10 BCP-ALL cases with *IGH-DUX4*. Authors reported that gc-HTS was a robust method for detecting *IGH*r requiring only a small amount of diagnostic material [40]. Further validation of this technique in a clinical setting is pending, though unless probes are designed to capture *DUX4*, this assay will miss *DUX4*r that does not involve *IGH*.

More commonly, research groups have relied upon transcriptome sequencing for analysis and detection of patients with *DUX4*r. Transcriptome sequencing allows for detection of transcribed gene fusions as well as GEP. Both direct detection of *DUX4*r through the identification of paired-end reads linking *IGH* and *DUX4* [9,12,13,14,17,18,19], as well as the detection of the distinct expression profile characteristic of this subtype [14,17,18,19], are possible with RNA sequencing (RNA-seq) data. A variety of different fusion calling algorithms have been employed for direct detection of *DUX4* and its fusion partner, including fusionCatcher [64], TopHat-fusion [65], defuse [66] and Cicero [67]. However, these algorithms do not consistently detect a fusion involving *DUX4* in all cases where a *DUX4*r has been indicated by GEP [14,18,19]. Alternatively, direct detection and comparison of expression levels of *DUX4* can be used to infer the presence of a *DUX4*r since *DUX4* is not expressed in healthy somatic tissue or leukaemic cells other than those with the rearrangement [12]. However, accurate quantification of *DUX4* expression requires consideration of the potential for short sequenced reads from the chimeric *DUX4* transcript to align to multiple locations in the human reference genome [9].

Given the highly distinct GEP observed in patients with *DUX4*r [6,8], hierarchical clustering of expression data can reliably identify patients with this subtype. This can be performed using gene sets reported in the literature as being associated with the *DUX4*r subtype [14], or alternatively by performing unsupervised hierarchical clustering utilising the top 5–10% of genes with the highest variability in expression across a large cohort of ALL samples [17,18,19]. In both instances, RNA-seq data from a large number of samples is typically required and is thus difficult to perform outside of a research context. To attempt to resolve this issue, Brown et al. [68] have developed a random forest classifier trained on RNA-seq data from two published cohorts for detection of *DUX4*r and other B-ALL subtypes. However, the classifier has not yet been tested on a large validation cohort containing multiple *DUX4*r samples.

## 6. Clinical Presentation 

*DUX4*r leukaemia has been reported in 4–7% of pediatric (<18 years) B-ALL patients [6,9,19], and has been associated with an older age of onset [13,47,48]. For example, Lilljebjörn et al, [13] report a median age of 6.5 years at time of diagnosis for paediatric *DUX4*r patients compared with 4 years for other B-ALL patients, while Marincervic-Zuniga et al, [15] report a median age of 9.3 compared to 4.5 years in their cohort. Studies investigating B-ALL in the adolescent and young adult population (16–39 years) suggest an enrichment in the proportion of *DUX4*r patients [12] although the lesion is observed across all age groups [17,18] (Table 1). At diagnosis, patients frequently present with lower white cell counts (WCC) compared with other B-ALL subtypes [16,48]. Immunophenotypic data indicates that *DUX4*r is exclusively associated with a B cell immunophenotype although aberrant expression of CD2, a T-lineage restricted cell marker, has been reported in a proportion of cases [7,48]. Patients have also been associated with lineage switching after the onset of treatment in which samples coexpress markers of both B lymphocyte and monocyte lineage. In addition, analysis of B-ALL samples which display lineage switching (swALL) show enrichment for *DUX4*r. This is observed as coexpression of CD19 and CD34 as well as CD33 and CD14. Not all *DUX4*r samples demonstrate swALL but those that do are reported to have poorer treatment response [63] and a higher rate of relapse [20] than non-swALL *DUX4*r.

## 7. Prognosis and Treatment 

As *DUX4*r is a recently described molecular lesion, there is currently no available data from prospective trials demonstrating its utility as an independent marker for risk stratification. This is hampered by difficulty in case ascertainment, as the lesion cannot be detected with standard diagnostic techniques. Patients with *DUX4*r have been reported to have higher levels of MRD throughout induction [14,40,48] and slower response to treatment [47]. Consequently, many patients within the *DUX4*r subtype are classified as intermediate or high risk based upon treatment response [14,40,47]. However, early reports investigating *DUX4*r subtype suggested that patients demonstrated excellent prognosis, particularly in paediatric patients who received intensive chemotherapy for remission induction [6,8]. 

Yasuda et al. [12] reported that patients with *DUX4r* demonstrated longer disease free survival after complete remission (CR) was achieved in adolescent and young adult patients; while Harvey et al. [8] reported significantly superior outcomes with 94% 4-year event free survival (EFS), though in the context of intensive chemotherapy given for high risk stratification (Table 1). Some studies have also reported good response to prednisolone in a majority of *DUX4*r cases [16,48]. Indeed the NALM6 cell line which demonstrates high expression of *PDGFRA*, a hallmark of ERG-deleted cases, was shown to be more sensitive to prednisolone than patients with low expression of *PDGFRA* [69].

Reports from more recent studies are varied and indicate differences in prognosis between pediatric and adult patients [19]. For example, in a study of 1988 B-ALL patients conducted by Gu et al. [17], *DUX4*r was associated with 93% EFS and overall survival (OS) in pediatric patients (<18) but 86% EFS and 84% OS in adults. Overall, it appears that presence of the fusion is associated with favourable outcomes, at least in the context of intensive chemotherapy applied as part of a risk-adapted approach to therapy. However, additional research on large uniformly treated cohorts is needed to confirm this finding. The reported protective effects of *ERG* deletion, and any other cooperating lesions, also requires further investigation to determine if this may enable stratification of patient risk within the *DUX4*r subtype [10,16]. 

Several B-ALL translocations, particularly those resulting in the production of chimeric kinases, have been shown to be targetable with precision medicine approaches [70]. *DUX4*r may constitute another B-ALL subtype which would be amenable to targeted therapy. Knock-down experiments utilising shRNA targeting the *DUX4* fusion transcript have been effective at reducing cell proliferation [12]. Furthermore, mutations targeting the homeodomains of *DUX4* can reduce DNA-binding activity of the transcription factor and abrogate the altered GEP of *DUX4*r [71]. This suggests that patients harbouring a *DUX4*r may be amenable to small molecule inhibitors that impede DNA binding. While there are currently no *DUX4* inhibitor molecules available or in clinical trials, pre-clinical validation of a potential *DUX4-IGH* inhibitor is underway that could potentially lower treatment toxicity associated with intensive chemotherapy [72]. 

## 8. Conclusions

*DUX4*r represents a distinct subtype of B-ALL affecting 4–7% of pediatric patients, and a higher proportion of adolescents and young adults. Patients typically exhibit favourable outcomes in the context of intensive chemotherapy, but conjecture does exist and additional studies in large cohorts of uniformly treated patients are needed to confirm this finding. To this end the development of a specific and sensitive assay for detection of either *DUX4* expression or the *DUX4*-fusion at diagnosis is essential. Early identification of patients with *DUX4*r as well as accurate identification of co-occurring alterations would enable studies to determine if secondary alterations such as *ERG* deletion are in themselves offering protective benefit, or if all patients with *DUX4*r subtype demonstrate good outcomes. Additional research into the mechanism of disease is also needed, specifically the impact of patient-specific variation in the DUX4-fusion and a greater understanding of the role of ERG deregulation in disease progression.

## Figures and Tables

**Figure 1 cancers-12-02815-f001:**
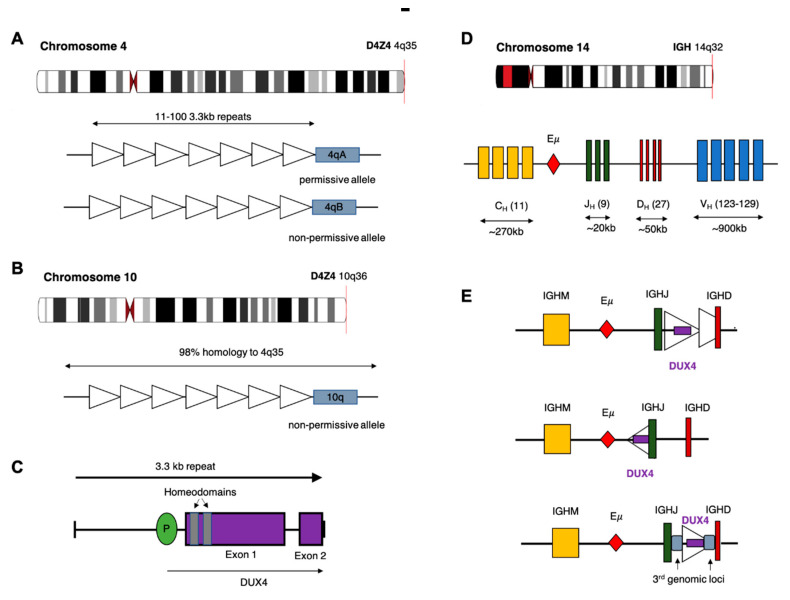
Potential chromosomal rearrangements involving *IGH* and *DUX4*. (**A**) Ideogram of chromosome 4 indicating location of the D4Z4 array and a depiction of the two alleles which vary in the sequence distal to the final repeat (repeat indicated by open triangles). This includes the permissive 4qA allele which can result in FSHD when contracted to fewer than 10 repeats. (**B**) Ideogram of chromosome 10 indicating location of the homologous D4Z4 array with 98% identical sequence. This chromosome is associated with a non-permissive allele which does not result in FSHD. (**C**) Schematic diagram of the repeat 3.3 kb repeat sequence indicating location and exons of the DUX4 gene. (**D**) Ideogram of chromosome 14 and depiction of the *IGH* locus indicating constant (C_H_), joining (J_H_), diversity (D_H_) and variable (V_H_) alleles (**E**) Schematic diagram depicting possible rearrangements as a result of cryptic insertion of *DUX4* from either chromosome 4 or 10 into the *IGH* locus. *DUX4* can be inserted in either orientation, include only a partial or one complete and one partial copy of the repeat array and also be inserted with sequence from a third genomic location.

**Figure 2 cancers-12-02815-f002:**
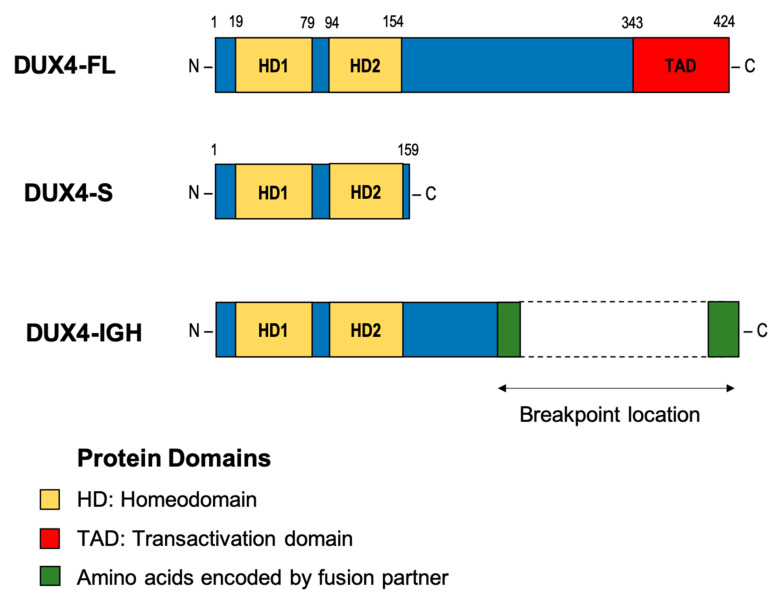
Deduced proteins of the various forms of DUX4. Full-length DUX4 (DUX4-FL) and truncated DUX4 (DUX4-S) generated from an alternatively spliced transcript of *DUX4* (*DUX4-s*) lacking the C-terminal transactivation domain. Putative DUX4-IGH chimeric protein in which the C-terminal portion of DUX4 containing the transactivation domain is replaced with amino acid sequence encoded by the genomic location into which *DUX4* was inserted. Length of the C-terminal portion of the chimeric protein can vary between patients, depending on the breakpoint location within exon 1 of *DUX4*.

**Table 1 cancers-12-02815-t001:** Summary of findings from published studies identifying *DUX4*r subtype.

Reference	Detection Method	Cohort (Age Range in Years)	Frequency *DUX4*r	Additional Alterations				Prognosis
				*ERG*	*IKZF1*	*PAX5*	*CDKN2A*	
Yeoh et al. 2002; [6]	Microarray GEP	Paediatric ALL (<19)	14/327 (4.3%)	—	—	—	—	—
Mullighan et al. 2007; [7]	Microarray GEP	B-ALL	19/218 (8.7%)	13/19 (68.4%)	—	—	—	—
Harvey et al. 2010; [8]	Microarray GEP	HR * B-ALL (1–20)	21/207 (10.1%)	8/21 (38.1%)	6/21 (28.6%)	3/21 (14.3%)	5/21 (23.8%)	4-yr RFS 94% ± 5.1
		HR * B-ALL (validation cohort)	5/99 (5.1%)	—	—	—	—	1/5 (20%) relapse
Zhang et al. 2016; [9]	Microarray GEP RNA-seq (n=175)	Paediatric B-ALL (0–15)	94/1347 (7.0%)	54/91 (59.3%)	17/91 (18.7%)	3/91 (3.3%)	—	—
		Adolescent B-ALL (16–20)	38/395 (9.62%)	23/38 (60.5%)	19/38 (50%)	6/38 (15.8%)	—	—
		Young Adult B-ALL (21–39)	9/171 (5.3%)	3/9 (33.3%)	3/9 (33.3%)	2/9 (22.2%)	—	—
Yasuda et al. 2016; [12]	RNA-seq	AYA Ph-negative B-ALL (15–39)	12/62 (19.4%)	—	—	—	—	8 CR; 1 CR after SCT; 1 Early mortality; 2 NA
Lilljebjörn et al. 2016; [13]	RNA-seq	Paediatric B-ALL (<18)	8/195 (4%)	5/8 (62.5%)	—	—	—	No observed relapses
		Paediatric B-Other ^ (2–15)	20/49 (40.8%)	10/20 (50%)	—	—	—	4/20 (20%) relapse
Liu et al. 2016; [19]	RNA-seq	Children (<18)	6/94 (6.4%)	—	3/6 (50%)	4/6 (66.7%)	3/6 (50%)	5-yr OS 100%
		Adult (>18)	5/78 (6.4%)	—	1/5 (20%)	3/5 (60%)	3/5 (60%)	5-yr OS 53%
Vendramini 2017; [16]	Microarray GEP	Paediatric B-Other ^ (<18)	35/143 (24.5%)	14/34 (41.2%)	12/34 (35.3%)	4/34 (11.8%)	5/34 (14.7%)	5-yr EFS 91.1% ± 4.9
Marincevic-Zuniga 2017 [15]	RNA-seq	Paediatric B-ALL (<18)	9/116 (7.8%)	7/9 (77.8%)	—	—	—	1/9 (11.1%) relapses
Li et al. 2018; [18]	RNA-seq	Children (<18)	50/906 (5.5%)	—	—	—	—	—
		Adult (>18)	13/258 (5.0%)	—	—	—	—	—
Zur Stadt et al. 2019; [40]	gc-HTS	B-ALL (excludes *ETV6-RUNX1*, *KMT2A*r and Ph+ALL)	10/164 (6.1%)	2/10 (20%)	—	—	—	—
Zaliova et al. 2019; [14]	RNA-seq	Paediatric B-Other ^ (1–18)	30/110 (27%)	19/30 (63%)	6/30 (20%)	6/30 (20%)	9/30 (30%)	—
Gu et al. 2019; [17]	RNA-seq	Paediatric (0.2–15)	61/1191 (5.1%)	—	2/35 (5.7%)	—	—	Child (<18) 5-yr EFS and OS 93.2% ± 3.8 Adult (>18) 5-yr EFS 84.6% ± 10; 5-yr OS 85.7% ± 9.4
		AYA B-ALL (16–39)	33/419 (7.9%)	—	3/18 (16.7%)	—	—	
		Adult B-ALL (40–79)	12/378 (3.2%)	—	—	—	—	

Abbreviations: GEP (gene expression profiling); RFS (relapse free survival); HR (high-risk); CR (complete remission); SCT (stem-cell transplant); EFS (event free survival); OS (overall survival).— Indicates data not provided in the cited study; * Determination of high-risk (HR) status based on high white cell count (WCC), older age of disease onset, and lack of favourable genetic features (*ETV6-RUNX1* or trisomy of chromosomes 4/10). ^ B-Other refers to B-ALL patients that do not belong to one of the following recognized subtypes: *ETV6-RUNX1*, *BCR-ABL1*, *TCF3-PBX1*, *KMT2A*r, hyperdiploidy or hypodiploidy.

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
