# Peer review of "DUX Hunting—Clinical Features and Diagnostic Challenges Associated with DUX4-Rearranged Leukaemia"

_cancers, 2020, doi:10.3390/cancers12102815_

Round 1

Reviewer 1 Report

This review focuses on DUX4-IGH BCP ALL subtype. The review highlights the difficulty to detect this subtype by routine techniques i.e. cytogenetics and PCR to, more importantly, properly stratify patients according to risk. The review is well structured, well referenced, is easy to read and images are clear and well described. Some minor comments (such as additional references) are described below:

  1. Introduction

- Lines 44-45: Authors must also include t(1;19) TCF3-PBX1.

- Lines 72-73: Authors say DUX4r cases show specific GEP. Authors could include that specific non coding RNA have also been described or just include references Vrendamini Oncotarget 2017 and Rani James Journal of Hematology & Oncology 2019. In addition, authors could mention that DUX4r shows its own methylation profile (Marincevic-Zuniga Journal of Hematology & Oncology 2017, Schroeder Sci Rep 2019).

  1. Disease mechanism

- Line 177: although it is referenced at the end of the sentence, authors could include ERGalt in the sentence to highlight this concept. Something like: “DUX4 has also been shown to induce expression of alternatively spliced transcripts (i.e. ERGalt) by binding to alternate transcription initiation sites [9,35]”.

  1. Genomic landscape of DUX4r leukemia

- Line 191: authors could include reference Taoudi Genes & Development 2011.

- Line 201: although it is highly dependent on the technique used, the frequency of ERG deletions in DUX4r ALL could to be around 80% by NGS (Zaliova Haematologica 2019). However, according to Table 1 it seems that ERG deletion is around 50%-60%. Therefore the frequency of DUX4r without ERG DEL would be around 40%. Authors could consider to increase up to 40% the percentage of non ERG DEL DUX4r in the sentence.

- Lines 209-210: authors suggest that low ERG expression correlates with lower proliferation. Authors could include, if they consider so, that low expression of ERG in AML (Nibourel Leukemia 2017) correlates with better prognosis and high ERG expression is seen in poor risk T-ALL (Baldus JCO 2007).

- Line 228: authors could include Li PNAS 2018 and Zaliova Haematologica 2019 in addition to the ref [9] at the end of the sentence.

  1. Detecting DUX4r

- Line 267 I think the ref [9] goes after the word “locus”.

- Line 317-318: there are repeated words in the same sentence.

- Line 327 heading of the section is 5 and should be 7.

- Line 365 heading of the section is 6 and should be 8.

- Prognosis and Treatment section:

Authors could mention data regarding the good response to prednisolone of DUX4r patients (Jerchel Haematologica 2017).

- References section:

Ref 61 is in CAPITAL letters.

Reviewer 2 Report

This is a timely review of DUX4-rearrangements in B-ALL with an introduction that covers some historical aspects, a detailed description of the rearrangements, putative mechanisms of leukaemogenicity, clinical aspects including continuing debates over their prognostic implications and a discussion of the challenges of routine testing to pick up these abnormalities. 

It is well-written and very readable with very few typographical errors. 

I have only a couple of very minor comments / suggestions:

1. In the clinical presentation section there is some description of cross-lineage aspects including co-expressioon of monocytoid or myeloid markers. It would be interesting to include some comment on whether these rearrangements have been described in any other haematological malignancies, particularly those involving monocytoid lineage eg. JMML, CMML. The information about varying incidence with age is a little brief - are there any data which could be included that include older adults with B-lineage ALL?

2. I felt that the ordering of the sections was slightly counter-intuitive. The 'diagnosis and treatment' section (currently mis-numbered as section 5, rather than section 7) would seem to better follow the 'clinical presentation' section. The 'detecting DUX4r' may better follow the earlier sections describing the structure of the rearrangments. I think this might improve the flow of the article, but is only a personal opinion!

Reviewer 3 Report

Line 283: clarify why CD371 and CD2 or RT-PCR testing has not been employed in clinical practice?

  • Please make sure to reference the recent B-ALL manuscript PMID32302940
  • Include an illustrative diagram of the rearrangement mechanism and altered gene expression of the DUX4 rearrangements

  • It is not clear what this manuscript is more focused on:  DUX4 disease mechanism or ability to detect it?

Reviewer 4 Report

The review by Rehn et al. is concise, comprehensive and well written. The authors consider different aspects of DUX4 expression in cancer cells.

Minor remarks.

1. In the introduction, the authors refer to the structure of the 4q35 locus, mostly known from studies on FSHD. In figure 1, I would avoid mentioning the pLAM sequence that is an  outdated concept, but rather discuss the 4qA/4qB polymorphism in more detail. 

2. Line 159 The phrase "This ‘Goldilocks’ level of expression hypothesis would suggest that the transcription of aberrant DUX4 is less important than the exact level of expression" is unclear and should be removed or clarified.

3. Is is noteworthy that the role of DUX4 in cancer has been suggested before the discovery of its involvement in BCP-ALL (PMID: 24341522) and that the implication of DUX4 in various cancers was recently reviewed by several authors. This fact may be worth mentioning in the text.

    4. Line 256 and below. The text is difficult to read. A figure can be helpful for the reader.

    Author Response

    Reviewer 5 Report

    “DUX hunting…”, a review by Rehn et al., is focused on the DUX4-rearrangement reported in a subset of B cell acute lymphoblastic leukemia (B-ALL) and covers clinical manifestation, genetics, and diagnostics.  The diagnostic section was excellent and built upon the prior presented data the authors arrived at the same conclusion I would have, using RNAseq gene expression clustering.  Overall, informative and generally will written and worthwhile with a few points that need clarification, outlined below:

    Conceptually hard to imagine a pathogenic coding function for the DUX4r since it is stated that multiple repeats can be inserted, and the repeat can be sense or antisense orientation (ref 12 and fig 1C).  Doesn’t this suggest a noncoding role?  This runs counter to what is stated in the following paragraph and in Fig 2 that say the PAH domains are always retained.  Proteins are described as deduced, but have they ever been detected? 

    Similarly, it is stated that the DUX4 TAD is replaced but the replacement sequence can be highly variable.  So is this an activator, repressor, dominant negative?  Does it matter?

    Figure 1 makes it seem only Chrom 4q35 D4Z4 are part of the rearrangements but the text indicates Chrom 10q26 are as well. Conceptually, they both should lead to the same thing as there is no coding difference, the difference is the distal PAS, chrom 4 has it (when permissive) and chrom 10 never has it (always nonpermissive).  Should add chrom 10 to the figure.

    When these rearrangements happen, do the patients get FSHD?  What happens to the chr 4q35 D4Z4 array?  Is it contracted?  Deleted?

    Top of page 5 is confusing.  First, the ‘Goldilocks’ bit hypothesizes that DUX4 goes into an epigenetically silenced allele when it starts in an epigenetically silenced allele and goes to an active allele upon recombination.  Second, they refer to DUX4 and not DUX4r yet in the prior Fig 2 they point out that these are always fusions of DUX4 without its TAD.  IF sometimes you can have full length DUX4 in these rearrangements then just need to fix the insertion FROM epigenetically silenced and INTO epigenetically active, and it makes sense.  The Yasuda model also makes sense as stated.  But why would these two very different models give the same clinical phenotype and GEP?  Or don’t they?  Needs clarification.

    In Section 4, the presented data seems to indicate these DUX4r B-ALL subtypes arise from a general genomic instability, and D4Z4 recombination is one of many instabilities.  Is the transformation all due to the DUX4r fusions or are there other transformative events or is it not known?

    The second section 5 at the end should be 7

    Pg 3 Contraction of D4Z4 leads to FSHD should ref van Deutekom 1993 and Wijmenga 1992.

    Author Response
